# Impact of Exposure to Indoor Air Chemicals on Health and the Progression of Building-Related Symptoms: A Case Report

Hiroko Nakaoka [1,2,*], Norimichi Suzuki [1], Akifumi Eguchi [1], Daisuke Matsuzawa [3] and Chisato Mori [1,2]

1 Centre for Preventive Medical Sciences, Chiba University, 1-33, Yayoicho, Inage-ku, Chiba 263-8522, Japan
2 Department of Bioenvironmental Medicine, Graduate School of Medicine, Chiba University, 1-8-1, Inohana, Chuo-ku, Chiba 260-8670, Japan
3 Research Center for Child Mental Development, Chiba University, 1-8-1, Inohana, Chuo-ku, Chiba 260-8670, Japan
* Correspondence: hnakaoka@faculty.chiba-u.jp; Tel.: +81-43-290-3896

**Abstract:** The aetiology of building-related symptoms (BRSs) is not well supported by sufficient scientific evidence, and it remains unclear whether BRSs are mediated by psychosocial and personal factors or a genuine physical susceptibility to low-dose chemical exposure. In April 2014, a 40-year-old man consulted the Environmental Medical Clinic at Chiba University complaining of recurring BRSs. Indoor air samples were collected from the patient's house at 11 time points and subjected to chemical analyses. The patient simultaneously completed a questionnaire about his symptoms at the time of the measurements. Statistical examination of the indoor environmental factors and patient survey revealed that the patient's symptoms were highly correlated with the indoor air quality. Additionally, ventilation may have mitigated his BRSs, whereas aerial odour did not trigger symptoms. These findings suggest that exposure to specific airborne chemicals in an indoor environment can cause BRSs, and ventilation may be one of the treatment options to mitigate symptoms. Additional investigations on the adverse impacts of airborne environmental chemicals on human health are necessary to develop effective treatments and establish preventive measures for BRSs, and further improvement of ventilation systems is required to ensure clean indoor air.

**Keywords:** building-related symptoms; chemical exposure; indoor air quality; social sustainability; volatile organic compounds





## 1. Introduction

Building-related symptoms (BRSs), also commonly known as sick building syndrome (SBS) symptoms, develop in an individual in association with a particular building and disappear or improve when the individual moves away from the building. BRSs are characterised by several symptoms linked to low-level exposure to multiple chemicals, primarily to volatile organic compounds (VOCs) in building materials, paints, furniture, pesticides, kerosene, or printing ink. BRSs vary in terms of manifestation and severity, ranging from mucosal irritation to neurological issues and fatigue. Occupants in buildings with indoor environment problems report an array of symptoms [1–4]; however, specific or obvious causes remain unclear [5], necessitating the study of risk factors for BRSs, including indoor environmental, social, personal, and cognitive factors [6–9]. In a previous study, we found that a high sum of VOCs (ΣVOCs), odour, and individual susceptibility were key risk factors for BRSs [10,11]. In other studies, moulds, bacteria, microbial VOCs (MVOCs), and semi-volatile organic compounds (SVOCs), such as plasticisers and flame retardants, have been implicated as important factors for the occurrence of BRSs [12,13]. Moreover, psychosocial and personal factors are also suspected to have strong links with BRSs [14–16]. For example, Ghaffarianhoseini et al. identified physical, biological, chemical, psychosocial, and other individual-scale parameters as the major contributors to BRSs [17]. Chemical odours are also possible triggers of BRSs among chemically sensitive individuals [9,18,19].

To date, several theories, such as those described above, have been posited as mechanisms behind the occurrence of BRSs; however, further studies, including the examination of various case studies, are needed to clarify the relationship between BRSs and environmental factors and address their symptoms.

In modern society, building operations, such as heating, cooling, air conditioning, and lighting, significantly contribute to energy consumption and greenhouse gas emissions [20]; therefore, sustainable buildings are required. One of the 17 sustainable development goals (SDGs) of the United Nations is the mitigation of climate change. Therefore, increasing energy efficiency and reducing energy-related carbon dioxide emissions in buildings through high airtightness and high heat insulation are recommended [21]. At the same time, human health is one of the SDGs, which requires fostering quality of life and human well-being, i.e., social sustainability [22,23].

In April 2014, a 40-year-old man visited the Environmental Medical Clinic, Chiba University, Chiba, Japan, complaining of recurring BRSs when he is at home; symptoms included concentration lapses, a heavy sensation in the head, and shoulder pain. This case study aimed to explain the aetiology of the patient's recurring symptoms and evaluate whether improving the indoor environment could alleviate and eliminate the BRSs. Furthermore, through this case study, we aimed to suggest the development of indoor environment designs that could establish environmental and social sustainability.

## 2. Materials and Methods

### 2.1. The Patient and Home Environment

The patient in this study was a medical doctor and researcher of psychiatry with no pre-existing psychiatric disorders. He complained of headaches and concentration lapses after installing an electronic piano at his house (Figure 1), and that the symptoms appeared when his child's toys, such as clay toys and game cards, and a children's book were left in the room. To investigate the relationship between the patient's symptoms and the indoor air environments, we asked the patient to remain in his home and report on the intensity of his symptoms whilst concentrations of indoor airborne chemicals were measured under various conditions. Between November 2014 and October 2015, indoor air samples were collected at 11 time points from the living room of the patient's house, where the piano was installed and where he had set up his university office and worked (Table 1). The patient reported no symptoms when he was outdoors; therefore, the outdoor environment air samples were used as a control. The indoor environmental conditions (e.g., temperature and humidity) were recorded during each sampling interval. A total of 78 chemicals were detected in the air samples and analysed.

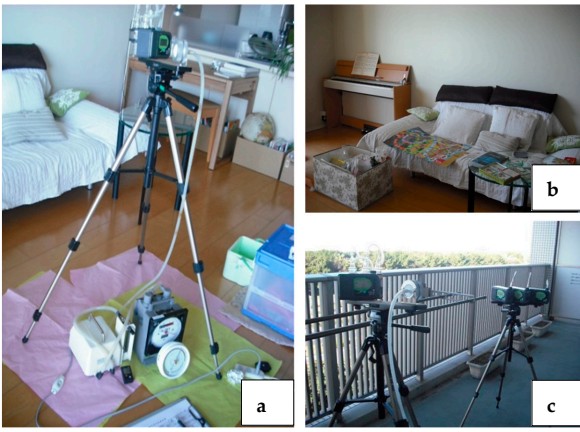

**Figure 1.** (**a**) Indoor air sampling in the patient's living room using the active sampling method. (**b**) Living room, where the electronic piano, clay toys, game cards, and a book for children were kept. (**c**) Outdoor air sampling using the active sampling method.

**Table 1.** Symptoms reported by the patient in indoor environments.

| ID | Situation | Y/M/D | ΣVOCs µg/m³ | Temperature °C | Humidity % | Symptoms Description | Symptom Score | Odour Score |
|---|---|---|---|---|---|---|---|---|
| 1 | LR * with EP **, a book for children | 2014.11.18 | 457 | 21.5 | 51.1 | Shoulder pain, Ear pain, Joint pain Tolerable for 30 min | 3 | 0 |
| 2 | LR with EP after 30 min of ventilation | 2014.11.18 | 301 | 20.9 | 39.3 | Tolerable for 30 min | 2 | 0 |
| 3 | LR with EP, floor heating system on | 2014.11.18 | 594 | 24.3 | 45.4 | Shoulder pain, Heavy headed, Nausea, Cognitive decline, Intolerable | 5 | 0 |
| 4 | LR with EP, clay toys, game cards, a book and floor heating system on | 2014.11.18 | 554 | 23.6 | 43.7 | Heavy headed, Intolerable longer than 5 min | 4 | 0 |
| 5 | LR with EP, air conditioner, humidifier on | 2014.11.26 | 517 | 25.4 | 64.3 | Shoulder pain, Eye pain, Cognitive decline | 3 | 0 |
| 6 | Staff lounge of the university | 2014.12.02 | 236 | 22.6 | 50.5 | Few symptoms | 1 | 0 |
| 7 | Outdoor air | 2014.12.19 | 47 | 6.4 | 39.2 | No symptoms | 0 | 0 |
| 8 | Office of the university | 2014.12.19 | 119 | 15.6 | 31.0 | Shoulder pain, Strong sleepiness | 3 | 0 |
| 9 | LR with EP | 2015.02.27 | 633 | 19.6 | 48.5 | Slight shoulder pain, Eye pain, Tolerable | 2 | 0 |
| 10 | LR without EP | 2015.03.03 | 388 | 19.7 | 41.7 | Few symptoms | 1 | 0 |
| 11 | LR with EP and a book | 2015.10.05 | 622 | 26.7 | 55.7 | Shoulder pain, Heavy headed, Cognitive decline, Nausea, Eye pain, Intolerable | 5 | 0 |

* LR: Living Room; ** EP: Electric Piano.

After the above experiments, we conducted an emission resource test to identify the source of the pollutants that may have caused the patient's symptoms. In this test, we measured the concentrations of VOCs on the surface areas of the interior materials and the electric piano, which are suspected to be contributing to his symptoms, because the concentration of VOCs is much higher near the surface areas than that in the air.

*2.2. Indoor Air Sampling and Analysis*

Indoor air samples from the patient's home and university office and outdoor air samples were collected under the following conditions:

1.  The electronic piano was present in the living room with the children's book.
2.  The electric piano was present in the living room without the children's book and after 30 min of ventilation (i.e., opening the windows).
3.  The floor heating system was switched on, and the electronic piano was present in the living room without the children's book.
4.  The electronic piano, clay toys, game cards, and children's book were present in the living room, and the floor heating system was switched on.
5.  The air conditioner and humidifier were switched on, and the electronic piano was present in the living room without the clay toys, game cards, and children's book.
6.  Staff lounge at the university.

7. Outdoor air was sampled as a control.
8. The patient's office at the university.
9. The electronic piano was kept inside the living room without the clay toys, game cards, and children's book.
10. The electronic piano was removed from the living room.
11. The electronic piano was present in the living room along with the children's book.

### 2.3. Analysis of Indoor Air and Emission Resource Test

Sixty-two VOCs and 16 aldehydes were analysed in the air samples according to the standard methods of air sampling and measurement issued by the Ministry of Health, Labour and Welfare (MHLW), Japan. All air samples were collected actively for 30 min. The Tenax® TA tubes (Supelco, Sigma-Aldrich Inc., Burlington, MA, USA) were used to capture VOCs, and DNPH Active Gas Tubes for Aldehyde (Shibata Science, Saitama, Japan) were used to capture aldehydes in the air samples. The air flow rates in the Tenax® TA tubes and DNPH Active Gas Tubes were set to 100 and 1000 mL/min, respectively. The collected VOCs were extracted by thermal desorption using the Turbo Matrix ATD (PerkinElmer, Waltham, MA, USA) and analysed using an Agilent 7890B gas chromatograph (Agilent Technologies Inc., Santa Clara, CA, USA) equipped with an MSD 5977A quadrupole mass spectrometer (Agilent Technologies, Inc., Santa Clara, CA, USA). Thermal desorption was performed by heating the Tenax TA tubes for 10 min at 280 °C, with a split ratio 20:1 and a transfer line temperature of 230 °C. Then, DB-1 columns (gas chromatograph analytical columns, 30 m × 0.25 mm i.d. with a film thickness of 1 μm; Agilent) were used to separate the samples. The column temperature was maintained at 35 °C for 5 min, increased to 240 °C at a rate of 10 °C/min, and then analysed in scan mode (m/z 40 to 350).

The collected aldehydes were extracted using solvents and analysed using high-performance liquid chromatography (HPLC) on the Prominence Ultra-Fast Liquid Chromatograph device with two LC-20AD liquid supply pumps, an SIL-20AC autosampler, and SPD M20A photodiode array detector (Shimadzu Co., Kyoto, Japan) using the Ascentis RP-Amide HPLC column (150 mm × 4.6 mm i.d., 2.7 μm column; Sigma-Aldrich). The column flow rate was set to 1.0 mL/min, and the mobile phases comprised 40–80% acetonitrile in water, with gradient elution mode for 90 min.

For the emission source test, the fixed areas were covered with aluminium foil, and a sampler with a large sampling rate was used to estimate the emission rate, assuming that all the emitted gas was collected. As samplers, VOC-TDs (Supelco, Sigma-Aldrich Inc., Burlington, MA, USA) were used to collect VOCs, and DNPH Passive Gas Tubes for Aldehyde (Shibata Science, Saitama, Japan) were used to collect aldehydes. The target areas for measurement were the floor, walls, and ceiling of the living room of the patient's home and the surface of the electronic piano (power on) where his symptoms were observed. Samplers placed in the target areas were covered with 12 × 12 cm thick aluminium foil and adhered to each other with tape (3 M mending tape); the samples were collected for 6 h. At the same time, a sampler was hung in the centre of the room to measure the airborne concentration in the indoor air. During collection, windows were left open, and a ventilation fan used to provide ventilation. Analysis methods were the same as above: heat desorption–gas chromatography/mass spectrometry (GC/MS) for VOCs and DNPH derivatisation–HPLC for aldehydes.

In this study, total VOCs (TVOCs) referred to the sum of the concentrations of compounds identified in the part of the chromatogram that ranged from n-hexane (C6) to n-hexadecane (C16), using the response factor of toluene. ΣVOCs were defined as TVOC values plus concentrations of 2-propanol, pentane, methyl acetate, 1-propanol, ethyl acetate, formaldehyde, acetaldehyde, acrolein, acetone, and propanal, each of which was identified by its individual response factor.

### 2.4. Evaluation by Human Sensory Perception

Before evaluating the indoor air quality in the patient's environment, we explained the procedure to the patient and obtained written informed consent. Subsequently, he completed the quick environmental exposure and sensitivity inventory (QEESI©) questionnaire, a common screening instrument for chemical sensitivity [24,25]. During the sampling, the patient remained in the room and completed a questionnaire about his specific symptoms, including symptom severity. He was not informed of the concentration levels of the VOCs detected during this study.

### 2.5. Statistical Analysis

Partial least squares (PLS) regression in the SIMCA 13.0 software (Umetrics, Umeå, Sweden) [26] was used to analyse the potential correlations between the patient's reported symptoms and the elements of the indoor air environment, including airborne chemicals detected in the indoor air samples, temperature, and humidity. Prior to the statistical analyses, all values were standardised using the equation

$$z = (x - \mu)/\sigma \tag{1}$$

where $\mu$ represents the mean value and $\sigma$ is the standard deviation of the variables. The robustness was evaluated using the residuals ($R^2$), and the model predictive ability parameter ($Q^2$) was determined using seven-fold cross-validation, wherein every seventh sample (i.e., one-seventh) was excluded from cross-validation until each sample had been excluded once. $Q^2$ was optimised using a backward stepwise method in which variables with small values (<0.8) of variable importance (VIP) were excluded. Chemicals with a VIP value > 1.5 were identified as potential air quality markers.

### 2.6. Medical Research Ethics

This study was approved by the Research Ethics Committee of the Graduate School of Medicine, Chiba University, Chiba, Japan (approval no. 1850).

### 3. Results

Table 1 presents the ΣVOC concentrations, temperature, and relative humidity, along with the symptom description, symptom score, and odour score reported by the patient in each sampling time point/situation. Throughout the study, under almost all conditions, the concentrations of 13 VOCs were lower than the guideline values set by the MHLW (Table 2) [27]. Among the eleven samples, the TVOCs of four samples, S3, S4, S9, and S10, were 520, 460, 550, and 530 µg/m³, respectively, which were all higher than the TVOC interim target value set by the MHLW, which is 400 µg/m³. The concentration data of chemical substances, including TVOC and ΣVOCs identified in this study, are shown in Supplementary Table S1.

The QEESI score and results of the evaluation using the sensitivity determination criteria reported by Hojo et al. [28,29] indicated that the patient was highly sensitive to chemicals. His symptom scores ranged from 0 (no symptoms) to 5 (severe symptoms) and varied across sampling conditions, with the most severe symptoms corresponding to comparatively high ΣVOC values. At those times, the patient reported that his symptoms were intolerable, and he was unable to remain in the living room. Moreover, he even claimed a decline in his ability to think. In contrast, the patient's symptoms were less aggravated or absent when he was at the university office or in the living room after ventilation, and there were no symptoms when he was outdoors. His symptoms tended to increase in severity with an increase in the ΣVOCs and temperature (Table 1).

**Table 2.** Comparison of VOC concentrations with 13 VOC guideline [a] values and TVOC.

| VOC (μg/m³) | Guideline Values | S1 | S2 | S3 | S4 | S5 | S6 | S7 | S8 | S9 | S10 | S11 |
|---|---|---|---|---|---|---|---|---|---|---|---|---|
| Tetradecane | 330 | 1.1 | ND | 1.1 | 1.2 | 1.4 | 1.6 | ND | ND | 1.2 | 1.3 | 2.3 |
| Ethylbenzene | 3800 | 2.7 | 6.2 | 1.4 | 1.4 | 1.4 | 1.8 | ND | 2.9 | 3.2 | 1.2 | 2.6 |
| Styrene | 220 | ND | ND | ND | ND | 1.1 | ND | ND | ND | 8.7 | 1.1 | 1.4 |
| Toluene | 260 | 11 | 31 | 5.3 | 5.3 | 8.6 | 3.1 | 3.7 | 4.6 | 11 | 7.3 | 6.2 |
| Xylene | 870 | 2.3 | 4.9 | 1.7 | 1.6 | 1.5 | 5.2 | ND | 8.7 | 2.1 | 1.3 | 2.2 |
| p-Dichlorobenzene | 240 | ND | ND | ND | ND | ND | 2.1 | ND | ND | ND | ND | ND |
| Formaldehyde | 100 | 43 | 20 | 40 | 44 | 74 | 29 | 2.7 | 7.2 | 39 | 31 | 41.0 |
| Acetaldehyde | 48 | 14 | 9.4 | 14 | 23 | 21 | 16 | 1.5 | 3.5 | 30 | 13 | 6.5 |
| Di-2-ethylhexyl phthalate DEHP | 120 | 0.1 | - | - | - | - | 0.2 | ND | 0.3 | - | - | - |
| Dibutyl phthalate | 220 | 0.2 | - | - | - | - | 0.7 | 0.2 | 0.6 | - | - | - |
| Diazinon | 0.29 | ND | - | - | - | - | ND | ND | ND | - | - | - |
| Chlorpyrifos | 1 | ND | - | - | - | - | ND | ND | ND | - | - | - |
| Fenobcarb | 33 | ND | - | - | - | - | ND | ND | ND | - | - | - |
| TVOC | 400 | 370 | 220 | 520 | 460 | 390 | 100 | 37 | 69 | 550 | 330 | 530 |

Unit: μg/m³, [a] source of guideline values: Ministry of Health, Labour, and Welfare of Japan; TVOC: total volatile organic compounds, ND = not detected, - (hyphen): not conducted, S: situation.

The regression equation ($R^2Y$ = 0.994 $Q^2$ = 0.912) predicted the correlations of symptom severity with each of the chemicals with high accuracy (Figure 2). The symptom scores were positively correlated with the concentrations of certain chemicals, such as dodecane, texanol, decamethylcyclopentasiloxane (D5), isododecane, and tridecane, but were negatively correlated with humidity (Figure 3).

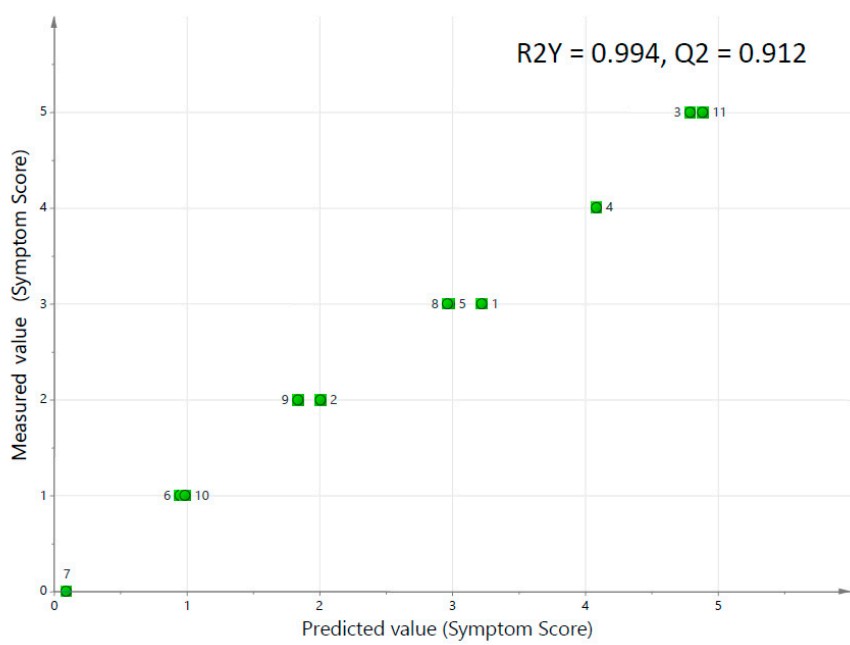

**Figure 2.** Predicted value and measured value. Y axis shows measured values, and X axis indicates predicted values. The correlation between symptom severity and chemical exposure was predicted with high accuracy using the regression equation ($R^2Y$ = 0.994 $Q^2$ = 0.912).

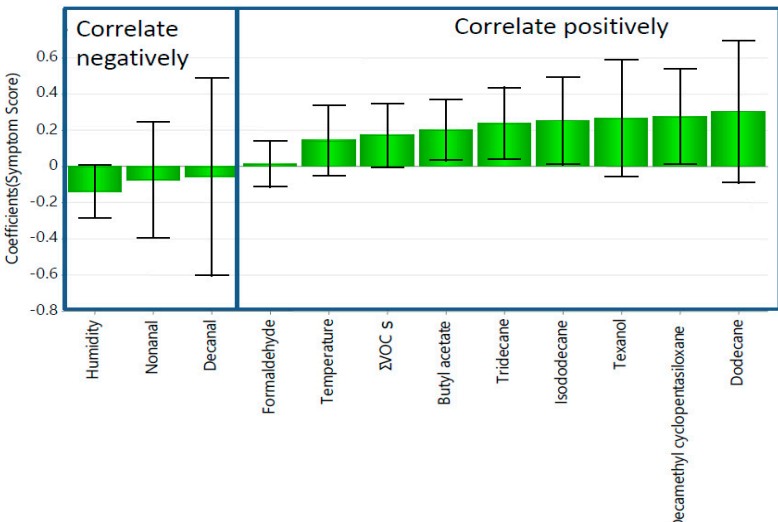

**Figure 3.** Correlation between symptom score and chemicals and environmental factors. Independent variables with longer bars have more influence on the dependent variable and symptom score.

The patient did not report any odours in any of the situations. The results of the emission source test are shown in Table 3. The average indoor temperature and relative humidity at the time of measurement were 28.6 °C and 70.3%, respectively. In the indoor air sampling conducted simultaneously, only isododecane and 2-ethyl-1-hexanol were detected at concentrations above 100 μg/m$^3$, while the concentrations of other substances were low. The emission rates of toluene and 2-ethyl-1-hexanol detected from the electronic piano were much higher than the concentrations of indoor air as a control.

**Table 3.** VOC concentration in indoor air and materials' surface areas in the living room of the patient's house.

| Compounds | Indoor Air | Surface of Interior Materials | | | |
|---|---|---|---|---|---|
| | | Ceiling | Wall | Floor | Electronic Piano |
| Isododecane | 151.4 | 95.8 | 119.6 | 33.5 | 141.5 |
| 2-Ethyl-1-hexanol | 141.1 | 74.8 | 69.9 | 80.9 | 212.5 |
| Limonene | 74.9 | 27.4 | 20.3 | 15.3 | 19.7 |
| Toluene | 59.7 | 62.4 | 40.3 | 40.7 | 409.3 |
| D5 | 22.6 | 18.3 | 16.4 | 15.1 | 13.7 |
| Formaldehyde | 16.2 | 4.4 | 6.0 | 1.7 | 3.7 |
| Acetaldehyde | 2.6 | 1.3 | 2.2 | 1.1 | 2.8 |

## 4. Discussion

BRSs are defined as medically unexplained symptoms attributed to possible exposure to environmental chemicals. The existence of BRSs is controversial; however, recent studies have suggested some legitimacy [30,31]. Accordingly, recent research has been devoted to determining the aetiology of these BRSs [32,33] and whether they are a result of exposure to airborne chemicals in the environment [34,35], misdiagnosis of other known physical [36] or psychosocial factors [37–39], or some combination of these [40]. In this study, concentrations of 78 VOCs were measured and analysed in the indoor air samples collected under 11 different sampling conditions. During air sampling, the patient remained indoors and completed the questionnaires about his symptoms. According to the responses obtained using the Japanese version of QEESI, the patient was categorised as vulnerable to chemical exposure. The investigation showed that the symptoms appeared in the patient even if the 13 VOCs were below the guideline values set by the MHLW and the TVOC values were lower or slightly higher than the interim target values. However, the symptoms

tended to increase in severity with increasing ΣVOC values and temperature, indicating that indoor air environment had a negative impact on the patient's health. His symptoms were alleviated or disappeared entirely after the living room was ventilated or when he was outdoors.

In a previous study, we found that ΣVOC values were significantly correlated with BRSs, especially among chemically sensitive individuals [10,11]. In this case study, we used PLS regression to statistically evaluate the correlations among the severity of the patient's symptoms, exposure to ΣVOCs, chemicals identified in the air, and environmental factors. This analysis revealed that the patient's symptoms occurred in various indoor situations and exhibited significant positive correlation with exposure to specific airborne chemicals, such as dodecane, texanol, decamethylcyclopentasiloxane (D5), isododecane, and tridecane; however, the patient's symptoms were negatively correlated with relative humidity. Therefore, the exposure to certain indoor airborne chemicals may have triggered the appearance of BRSs. There was a negative correlation between relative humidity and symptom severity, which is consistent with previous studies that reported that high relative humidity might alleviate sensory irritation [41,42].

In the emission source test, only the concentrations of toluene and 2-ethyl-1-hexanol emitted from the electronic piano were higher than that in the indoor air, suggesting that these emissions came from the surface of the electronic piano. Patient symptom scores were significantly correlated with the detection of isododecane, dodecane, and D5, but their sources could not be determined.

Although many studies have suggested a potential link between odour and BRSs [43–45], the patient in this study reported no odours under all conditions. In other words, while he reacted to some specific chemicals, he did not smell any odours, possibly because of the low doses of the airborne chemicals, for which the odour thresholds are relatively high (decane, 5030; texanol, 1327; isododecanae, 31347; tridecane, 905 μg/m$^3$; and D5, unknown) [46]. Rosenkranz and Cunningham (2003) have concluded that odour perceptibility does not have an impact on health [47]; the present findings also suggest that olfactory sensitisation or perception is not necessarily a trigger of BRSs.

In this study, we examined the relationship between BRSs and airborne environmental chemicals under various conditions in cooperation with a chemically sensitive patient. Although some investigators have previously reported an association of BRSs with psychosocial factors and social determinants [7,8,14,48,49], our findings suggest that exposure to specific airborne chemicals in the indoor environment was the probable cause of BRS onset. Furthermore, the results indicated that chemical concentrations can be used to predict the possible adverse effects of indoor air quality on human health and improve indoor air quality, especially for chemically sensitive individuals.

Post these experiments, the patient's symptoms have improved as follows.

1.    2014–2016

During this period, the patient's son was young, and his toys and books were always in the living room, and the electronic piano was there as well. Therefore, the patient's symptoms were rather severe. Thus, he stayed in the living room only for a short period (approximately one hour at breakfast and dinner each day), and the ventilation was strong during his stay. He spent most of his time in his room or office.

2.    2017–2018

When the patient's son entered the third grade of elementary school, his toys and books were moved from the living room to his room. The electronic piano still remained in the living room. The patient's symptoms became less severe but still appeared.

3.    2019–2022 (Present)

The electronic piano, which was suspected to be the main cause of patient's symptoms, has remained in the living room, but perhaps the emission of chemical substances has

decreased over time, and the patient's symptoms have almost disappeared even in the living room. The symptoms disappear even if he does not focus on ventilation.

4.    Family members

The patient's wife and son did not develop symptoms of BRSs despite living in the same environment.

In this study, the onset of the patient's BRSs was suspected to be caused by chemicals. Thus, it is necessary to first reduce the concentration of chemical substances in the environment. However, it took approximately five years after the patient complained of symptoms and consulted the Environmental Medical Clinic, Chiba University, before he was able to be comfortable in his living room without concerns about his symptoms.

Thus, to alleviate symptoms, individuals who are considered vulnerable to chemicals should avoid substances or places suspected of causing symptoms and ensure that the areas are well ventilated.

Furthermore, ageing also seems to reduce chemical emissions as they decrease over time. Using used items instead of new ones, such as furniture and household goods, may be an option for preventing BRSs.

The results of this study suggest the following two hypotheses regarding the mechanisms of BRSs. The first hypothesis is that metabolic enzymes in the patient may be deficient or weakly functioning. The fact that only the patient developed symptoms in the same environment as other family members suggests that the concentration of inhaled or dermal exposure to the chemicals may have been maintained in his body at relatively high levels. Henceforth, we plan to examine the correlation between blood or urine metabolite concentrations and symptom severity after environmental exposure to verify this hypothesis. Second, the symptoms of BRSs, such as exhaustion, mental instability, cognitive decline, and impaired concentration, seem to be widely associated with psychiatric disorders, particularly depression and other mood disorders. The similarity to the symptoms of psychiatric disorders suggests that a similar condition may be occurring in BRSs. A lack of neurotransmitters or an imbalance of neurotransmitters may be affecting the BRSs. It may be possible to develop drug-based mitigation measures in the future by examining the relationship between drug-induced changes in blood levels of neurotransmitters and the severity of symptoms.

Currently, reducing the concentrations of indoor air chemicals can reduce the occurrence of BRSs. The patient's symptoms were alleviated by avoiding exposure to chemicals and improving indoor ventilation; therefore, installing sufficient ventilation equipment may be an effective treatment for BRSs. Modern architecture limits natural ventilation to save energy by improving airtightness and heat insulation to achieve sustainable development goals (SDGs) [50]. However, it can potentially reduce ventilation and increase the concentration of indoor air pollutants. To prevent adverse health effects as a result of poor indoor air environment and ensure human well-being, new technology is needed to balance energy conservation and good indoor air quality [51].

In the midst of the COVID-19 pandemic, many studies have reconsidered the usefulness of ventilation to avoid airborne transmission of pathogens [52–54]. In addition, ventilation is essential for reducing indoor concentrations of air pollutants, such as VOCs, to prevent diseases. Morawska et al. (2021) argued that indoor air must be free of pollutants for the well-being of occupants, just as we expect with potable water from our taps [55]. Moreover, improvement of building ventilation systems can help with the mitigation and treatment of BRSs.

Environmental toxicology investigations are needed to prevent BRSs. In the future, we will conduct further emission testing to identify the sources of chemical emissions and confirm whether the elimination of these chemicals can effectively treat and prevent BRSs. In addition, we plan to verify our hypothesis regarding the mechanism of BRSs to develop appropriate treatments and preventive measures for BRSs.

One of the study's limitations is that there was only one patient who was considered to be more sensitive to chemicals than the general population. Furthermore, there are various

types of chemical substances present in indoor air with varying composition depending on the location, making it difficult to use the data of this study as a representation of population exposure. Additionally, indoor air samples were collected and analysed under 11 different conditions; this is considered insufficient as there are various chemical substances present in indoor air that were not sampled for. However, we conducted investigations to determine the change in severity of symptoms by varying environmental factors, while keeping the participant's characters and sensitivities the same. Despite these limitations, this case study statistically established that increasing concentrations of chemicals in indoor air could increase symptom severity. Another limitation is the difficulty of excluding the participant's subjective observation and biases since he is a psychiatrist/researcher and has original opinions about the cause of the symptoms.

The strength of this study was that participants stayed on site for the duration of the investigation and symptom severity scoring under the 11 different conditions was carried out simultaneously during chemical sampling and analysis. As a result, it was found that chemical substances triggered the onset of symptoms and that symptom severity varied depending on the concentration of a chemical substance. From these findings, we could infer the aetiology of symptom onset, which can influence future research to elucidate the mechanism. Regarding the limitation of the patient's vulnerability, the environment in which he is comfortable also means the environment that is comfortable for those who are not vulnerable. The results of this study are a necessary finding and another strength for carrying out environmental normalisation.

## 5. Conclusions

The patient likely developed BRSs from exposure to specific airborne chemicals present in the indoor environment. These symptoms appeared even at VOC levels below the guideline values set by the Japanese MHLW. Our findings suggest that certain non-regulated chemicals may cause BRSs even at low concentrations. Given the limitations of analyses of chemicals present at low concentrations, $\Sigma$VOCs could be used as markers to predict the effects of indoor air quality on human health. Despite controversy regarding the causes or aetiology of BRSs, investigations of the adverse impacts of airborne environmental chemicals are warranted to develop effective treatments and establish preventive measures for BRSs. Furthermore, the results of this study suggest that ventilation may be one of the treatment options to alleviate or eliminate BRSs. It can also be concluded that further improvement of ventilation systems to ensure clean indoor air is needed.

**Supplementary Materials:** The following supporting information can be downloaded at: https://www.mdpi.com/article/10.3390/su142114421/s1, Table S1: All the concentration data of chemical substances including TVOC and $\Sigma$VOCs identified in this study.

**Author Contributions:** Conceptualisation, H.N., N.S. and D.M.; methodology, H.N. and N.S.; formal analysis, A.E.; investigation, H.N., N.S. and D.M.; resources, D.M.; data curation, H.N., N.S. and D.M.; writing—original draft preparation, H.N.; writing—review and editing, H.N., N.S. and D.M.; supervision, C.M.; project administration, H.N.; funding acquisition, H.N. All authors have read and agreed to the published version of the manuscript.

**Funding:** This study was supported by Grants for Scientific Research (C): Grants-in-Aid for Scientific Research <KAKENHI (24510112)> by the Japanese Ministry of Education Culture, Sports, Science and Technology.

**Institutional Review Board Statement:** The study was conducted in accordance with the Declaration of Helsinki and approved by the Research Ethics Committee of the Graduate School of Medicine, Chiba University (approval no. 1850).

**Informed Consent Statement:** Written informed consent was obtained from the patient involved in the study.

**Data Availability Statement:** Not applicable.

**Conflicts of Interest:** The authors declare no conflict of interest.

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
