# Peer review of "Impact of Exposure to Indoor Air Chemicals on Health and the Progression of Building-Related Symptoms: A Case Report"

_sustainability, doi:10.3390/su142114421_

Round 1

Reviewer 1 Report

This paper did a case study on a patient for building related symptoms possibly due to indoor air chemical exposure. The following problems need to be addressed.

1. This research work was done in 2014 and 2015, which is 7 years ago. What is the situation from then on? It should be much more valuable. I would reject this paper if no new information can be provided.

2. It seems that the patient had a child. Were his family members investigated as well? This would be beneficial to this study.

3. What is the difference between the TVOC in Table 1 (4th column) and Table 2 (last row). It seems that the values have big differences.

4. In Table 2, the TVOC value is much higher than the total value of the individual VOC. What are the main VOCs? The authors should identify.

5. No deep analyses is provided in “Discussion“.

Author Response

Thank you for your comments. 

Regarding our replies, please see the attachment.

Reviewer 2 Report

The paper deals with the evaluation of Building-related symptomps (BRSs) of a 40 years old subject.

The manuscript cannot be considered a scientific paper, but rather a description of a specific case.

In the present form the paper is not sufficiently solid from a scientific point of view. The results are related to the experience of a single patient.

Although the individual experienced symptoms, these cannot directly linked to the presence of one or more pollutants, nor can the sample be considered significant for bringing out scientific evidence. Elements of subjectivity and BIAS also linked to the patient's profession cannot be excluded.

Although the topic is interesting, and a measurement activity has been conducted, the paper is not enought robust to be considered a research article.

Author Response

Thank you for your comments.

Regarding our reply, please see the attachment.

Reviewer 3 Report

The present study clearly and directly assesses the problem associated with negative health effects on people indoors, related to air quality. Nonetheless, The study evaluated only one patient. How the authors justify their conclusions regarding this scenario?

Figure 1 - Since there are three photographs in Figure 1, it is desirable that they all be numbered as1a, 1b and 1c

Author Response

Thank you for your comments and advice.

Round 2

Reviewer 1 Report

I don't think the paper is significantly improved regarding addressing the problems.

Author Response

Thank you for the comment.

We have revised the manuscript according to your previous comment and added the progress of the patient’s status after 2014 and our opinions in the Discussion.

Reviewer 2 Report

The authors worked on the paper, but in my opinion they have not solved the major problems relating to the significance of the obtained results.

I suggest at least adding a "Limitations" section in which to explain in detail all the limitations and weaknesses of the paper in order to clarify the real strength of the obtained results.

Author Response

Thank you for your comment.

As per the suggestion, we added two paragraphs to provide the limitations and strengths of the study under the Discussion section of the manuscript (lines 374–401).

Round 3

Reviewer 2 Report

The authors have included a part related to the limitations of the study as requested by me. I believe that the paper is now ready for publication.